# Epidemiology and Risk Factors of Carbapenemase-Producing *Enterobacteriaceae* Acquisition and Colonization at a Korean Hospital over 1 Year

**DOI:** 10.3390/antibiotics12040759

**Published:** 2023-04-14

**Authors:** Hye-Jin Kim, JungHee Hyun, Hyo-Seon Jeong, Yeon-Kyeng Lee

**Affiliations:** Division of Healthcare Associated Infection Control, Bureau of Healthcare Safety and Immunization, Korea Disease Control and Prevention Agency (KDCA), Heungdeok-gu, Cheongju-si 28159, Republic of Korea; hjjkim@korea.kr (H.-J.K.);

**Keywords:** acquisition, carbapenemase-producing *Enterobacteriaceae*, colonization, risk factors

## Abstract

**Background**: Carbapenemase-producing *Enterobacteriaceae* (CPE) are known to be primarily responsible for the increasing spread of carbapenem-resistant *Enterobacteriaceae* and have therefore been targeted for preventing transmission and appropriate treatment. This study aimed to describe the clinical and epidemiological characteristics and risk factors of CPE infection in terms of acquisition and colonization. **Methods**: We examined patients’ hospital data, including active screening on patients’ admission and in intensive care units (ICUs). We identified risk factors for CPE acquisition by comparing the clinical and epidemiological data of CPE-positive patients between colonization and acquisition groups. Results: A total of 77 CPE patients were included (51 colonized and 26 acquired). The most frequent *Enterobacteriaceae* species was *Klebsiella pneumoniae*. Among CPE-colonized patients, 80.4% had a hospitalization history within 3 months. CPE acquisition was significantly associated with treatment in an ICU [adjusted odds ratio (aOR): 46.72, 95% confidence interval (CI): 5.08–430.09] and holding a gastrointestinal tube (aOR: 12.70, 95% CI: 2.61–61.84). **Conclusions**: CPE acquisition was significantly associated with ICU stay, open wounds, holding catheters or tubes, and antibiotic treatment. Active CPE screening should be implemented on admission and periodically for high-risk patients.

## 1. Introduction

Carbapenem-resistant *Enterobacteriaceae* (CRE) infection exhibits resistance to one or more carbapenem antibiotics (doripenem, imipenem, meropenem, ertapenem), as reported by the Clinical and Laboratory Standards Institute (CLSI) (M100-30th ed, 2020). According to this guideline, doripenem, imipenem, and meropenem are considered resistant at a minimum inhibitory concentration (MIC) of ≥4, and ertapenem is considered resistant at an MIC of ≥2 [1,2]. CRE is becoming increasingly prevalent in healthcare environments worldwide and is emerging as a major public health concern [1]. Invasive infections caused by CRE are associated with high mortality rates (up to 40–50%, as reported by some studies), and in addition to β-lactam/carbapenem resistance, CRE often carries genes that confer high levels of resistance to many other antimicrobials, making treatment options extremely limited [3,4,5]. Carbapenemase-producing *Enterobacteriaceae* (CPE) produces enzymes that degrade carbapenems and easily transmit genes to other bacteria through mobile plasmids, resulting in a high risk of cluster outbreaks at healthcare facilities [6]. Therefore, CPE is considered to be of greater concern in terms of both infection prevention and treatment. Rapid detection of CPE is important for the timely implementation of appropriate prevention measures and determination of treatment regimens [1]. Based on their amino acid sequences, carbapenemases are classified into Ambler class A (serine carbapenemases), class B (metallo-β-lactamases), and class D (oxacillinase carbapenemases). The most common and problematic carbapenemases in Korea are *Klebsiella pneumoniae* carbapenemase (KPC, class A), imipenemase, Verona integron-encoded metallo-β-lactamase, and New Delhi metallo-β-lactamase (imipenemase [IMP], Verona integron-encoded metallo-β-lactamase [VIM], New Delhi metallo-β-lactamase [NDM], class B), and oxacillinase-48 (OXA-48, class D) [7].

To date, nursing home admission, ICU admission, invasive procedures such as nasogastric feeding tube placement and catheterization, recent antibiotic use, and prior antimicrobial exposure have been reported as independent risk factors for CPE carriage [4,8,9,10]. The U.S. Healthcare Infection Control Practices Advisory Committee (HICPAC) recommends periodical screening of stool or rectal swab samples to diagnose multidrug-resistant bacterial colonization, given that appropriate application of isolation measures through rapid diagnosis can prevent the transmission of contact infections, such as multidrug-resistant bacterial infection. The HICPAC also recommends staff training on medical infection characteristics and transmission routes [11,12]. Identifying the risk factors for CPE infections can help establish and improve CPE screening criteria for infection control of hospitalized patients [13,14]. In Korea, all patients with CRE/CPE infections are required to report to a notifiable diseases surveillance system managed by the Korea Disease Control and Prevention Agency (KDCA). In addition, active surveillance of cultures of samples, such as those of stool, rectal, skin sites, and wounds, is recommended to patients admitted with CRE risk factors or those admitted to risk settings (e.g., intensive care units) at admission and periodically (e.g., weekly) during hospital stay [2]. This study aimed to analyze the data of patients with CPE infection at a hospital with active surveillance and to describe the epidemiological and clinical characteristics and risk factors of CPE infection in terms of acquisition and colonization.

## 2. Results

### 2.1. General Characteristics

Among 77 CPE-positive patients, 66.2% were identified with CPE colonization on admission and 33.8% showed hospital-acquired CPE. The mean age was 80.0 years in the CPE colonization group and 71.5 years in the CPE acquisition group. Patients with hospital-acquired CPE were significantly older than those with CPE colonization on admission (*p* = 0.036). Furthermore, 54 of the 77 patients (70.1%) had a hospitalization history at other hospitals within 3 months; among these, 41 (75.9%) were confirmed to have CPE colonization on admission. Of 51 patients with CPE colonization, 49 (96.1%) were admitted to the emergency room (*n* = 25) and general ward (*n* = 24). Of 26 patients with CPE acquisition, 50% had a history of hospitalization at other hospitals within 3 months, and 42.3% were admitted to the ward (Table 1).

### 2.2. CPE-Related Characteristics

Data on CPE-related characteristics were isolated from rectal swab samples (including stools) from 72.5% of patients on admission (colonization) and 57.7% during hospitalization (hospital-acquired), respectively. The most common CPE species isolated were *Klebsiella pneumonia* and *Escherichia coli* in 60 and 14 patients, respectively. Two cases of *Citrobacter freundii* and one of *Citrobacter amalonaticus* were identified only in the CPE acquisition group.

KPC-producing bacteria were the most frequently isolated bacteria in 69 patients (89.6%). In addition, NDM-producing bacteria were isolated in five patients, OXA-48-producing bacteria in one patient, and both NDM and OXA-48-producing bacteria in two patients (Table 1).

### 2.3. Clinical Risk Factors

The average length of hospital stay until CPE acquisition was 16.8 days, and 80.8% of patients with hospital-acquired CPE received treatment in the ICU (*p* < 0.001). The incidence of invasive treatments with urinary catheters, central catheters, and gastrointestinal tubes (*p* = 0.038, *p* < 0.001, and *p* = 0.007, respectively) was significantly different between the two groups. Furthermore, mechanical ventilation (*p* = 0.026), exposure to carbapenems (*p* = 0.027), and glycopeptides (*p* = 0.006) were also significantly different between the two groups (Table 2).

### 2.4. Risk Factors for Hospital-Acquired CPE

Multiple regression analysis was conducted to identify the risk factors associated with CPE acquisition during hospitalization compared with CPE colonization at admission. Univariate analysis revealed that the risk of newly acquired CPE was lower by 0.32 times for patients aged ≥ 70 years than for patients aged < 70 years (*p* = 0.029); however, this difference was not statistically significant after adjusting for general characteristics (such as age, sex, hospitalization history at other hospitals within 3 months, admission route, and comorbidity). The risk factors for hospital-acquired CPE were identified in a multivariate analysis after adjusting for the general characteristics of the study population. The risk of CPE acquisition was 0.32 times higher when hospitalization history at other hospitals was within 3 months, which translates to a 3.12 times higher risk of having a CPE colonization on admission with a history of other hospitals within 3 months. The risk of CPE acquisition during hospitalization was 46.72 (*p* < 0.001) and 5.99 (*p* = 0.011) times higher in patients treated in the ICU and those with open wounds, respectively. Among the invasive treatments performed during hospitalization, urinary catheters, central catheters, and gastrointestinal tubes were 4.56 (*p* = 0.033), 11.94 (*p* < 0.001), and 12.7 (*p* = 0.002) times more likely to be associated with CPE acquisition, respectively. Furthermore, administering carbapenems and glycopeptides increased the risk of acquiring CPE by 4.88 (*p* = 0.013) and 5.13 (*p* = 0.011) times, respectively (Table 3).

## 3. Discussion

This study analyzed the characteristics and risk factors between patients with newly acquired CPE during hospitalization and those with CPE colonization on admission using the data of CPE infection reported through active surveillance cultures. Of the 51 patients identified with CPE colonization, 80.4% were identified to have a hospitalization history within 3 months and 96.1% were admitted through the emergency room and general wards. To prevent CPE cross-transmission in hospital settings from an external carrier, studies have emphasized the active screening of patients on admission and in health risk settings, such as ICUs [11,15]. In our study, 82.4% of patients with CPE colonization were screened in the emergency room and general wards, where active screening tests were not applied. This suggests that more patients with CPE colonization can be identified if comprehensive screening tests are performed for patients admitted to general wards or the ER. Similar to our study, Salomao et al. [16,17] show that screening high-risk patients on admission to the emergency department suggests a strategy for early identification of CRE carriage on admission to ER to control CRE transmission. Our study also highlights the need for screening tests for patients with CPE colonization status on admission to a general ward or emergency room as well as the ICU. However, considering the burden on healthcare facilities in terms of time and cost, risk-based screening has been used to limit the spread of CRE (CPE) in healthcare settings [18,19]. In addition, Kang and Jeong recommended that inpatients identified with a hospitalization history of other hospitals within 3 months should be screened first [20]. Of the 26 patients classified as acquiring CPE during hospitalization, there were 13 patients with a hospitalization history at other hospitals within 3 months of active surveillance cultures on admission, and 7 patients were first admitted to the ICU, with 2 patients having overlapping hospitalization history at other hospitals within 3 months and first admission to the ICU. Therefore, a total of 18 patients had undetectable CPE status on admission and acquired new CPE during hospitalization. Of the eight patients for whom active surveillance cultures were not performed because it was not their first ICU admission or because their hospitalization history at other hospitals within 3 months was unknown, seven were identified on routine weekly testing in the ICU (six detected at least 2 weeks after ICU admission, and one detected 6 days after ICU admission), and one was identified after ICU admission and general ward transfer, supporting the acquisition of CPE during hospitalization. The 26 patients with undetected CPE status on admission and new CPE acquisition during hospitalization were used to analyze the risk factors associated with CPE acquisition among therapeutic practices applied during hospitalization. Han et al. [13] identified ventilator use, the use of three or more antibiotics during an ICU stay, and high disease severity as risk factors for the acquisition of multidrug-resistant bacteria. Lee et al. [9] found that ICU admission in the last three months, antibiotic use, tracheotomy and endoscopy, central line use, ventilator use, nasogastric tube use, surgery, and catheterization were associated with increased risk of CRE acquisition. Similar to the above findings, the findings of the present study revealed the risk factors associated with CPE: ICU admission, ICU admission within 3 months, open wound, urinary catheter, central catheter, gastrointestinal tube, and use of carbapenem and glycopeptide antibiotics. As a result, we sought to prevent hospital-acquired CPE and implement active infection control through regular surveillance cultures for high-risk groups corresponding to the risk factors of CPE acquisition.

This study has several limitations. First, it was based on data from a single hospital. However, the hospital conducted active surveillance cultures for high-risk patients on admission and periodically among patients during their ICU stay. This allowed the comparison of epidemiological and clinical characteristics between the CPE colonization group and CPE acquisition group. Second, this study did not include direct and indirect factors associated with healthcare workers. Due to a single relatively short period, it was likely to be consistent. Third, the number of patients with CPE infection was possibly underestimated or overestimated because active surveillance culture was not conducted outside the ICU and patients without hospital history within 3 months. Furthermore, the total number of samples for CPE screening was not collected, and the CPE acquisition rate could not be calculated.

Despite these limitations, this study includes all CPE-positive patients over 1 year at the study hospital. CPE screening tests were relatively well implemented, with epidemiological investigation conducted to identify a range of in-depth risk factors in CPE-positive patients. In future research, the study should be expanded to multiple institutions, including a comparison of the resistance patterns of CPE strains by year, in order to increase the generalizability of the research results. Moreover, to achieve a better understanding of the causal relationship between the risk factors associated with CPE acquisition and comparison, in-depth research should be conducted that includes comparison groups, such as non-CPE.

## 4. Materials and Methods

To confirm the presence of CRE, medical institutions conduct their own identification and antimicrobial susceptibility testing of intestinal bacterial species according to the CLSI guidelines (M100-30, 2020) provided by the KDCA. If the intestinal bacterial species isolated from clinical samples meet the criteria for detecting carbapenem resistance, it must be reported to the KDCA Disease Surveillance System within 24 h. When CRE is identified, genetic testing must be performed to confirm the presence of CPE, and additional reporting is required. The results of CPE genetic testing typically require about a week.

A CPE monitoring program was conducted at the study hospital; a secondary general hospital with approximately 670 beds. Patients eligible for the CPE monitoring program were as follows: patients with hospitalization history at other hospitals within 3 months or patients admitted to the ICU who had undergone CPE testing at the time of admission. In addition, regular CPE screening was performed weekly in the ICU.

The study population included patients with CPE-positive results reported to the KDCA from 1 January to 31 December 2021, Among a total of 81 patients with CPE, we excluded one vulnerable patient aged <18 years and three patients in whom acquisition or colonization could not be identified. Ultimately, 77 patients with CPE were included in the study.

### 4.1. Data Collection

We used the study hospital’s CPE reports from the KDCA CRE surveillance dataset. The reports included: patients’ demographic information (sex, age) and CPE-related information such as the date of CPE isolation; type of carbapenemase; history of ICU treatment; surgery (excluding simple surgery); and exposure to antibiotics and clinical information, such as admission date, name of the isolated bacteria, sample type, sample collection date, route of movement within the hospital, type of surgery, and invasive treatment. Additional data, such as underlying disease, presence of open wounds, hospitalization history at other hospitals within 3 months, and use of drainage tubes, were collected from the medical records. We reviewed and confirmed all the information by matching the registered and case report information.

### 4.2. Definitions

A patient was described as being CPE colonized if they had CPE-positive culture findings from (i) a sample (stool or rectal swab) collected within 24 h of admission in terms of a CPE surveillance program conducted for patients who were either admitted to the ICU or those who had a hospitalization history at other hospitals within 3 months and (ii) clinical specimen culture of samples collected within 48 h (2 days) of admission. In contrast, patients with CPE-negative culture findings from the initial sample collected on admission but CPE-positive culture findings from active surveillance were described as being CPE acquired [12,21].

### 4.3. Data Analysis Method

Demographic and clinical characteristics of CPE-positive patients are presented as frequencies (percentage) or mean (standard deviation). The *t*-test and Fisher exact test were used to evaluate differences in demographic and clinical characteristics between patients with CPE colonization on admission and those with hospital-acquired CPE, depending on the characteristic. To analyze the relationship between clinical characteristics and CPE acquisition, the odds ratio (OR) of each variable was obtained using univariate logistic regression. Additionally, adjusted odds ratios (aORs) were analyzed using multivariate regression analysis after adjusting for age, sex, hospitalization history at other hospitals within 3 months, admission route, and comorbidity. To analyze risk factors of CPE acquisition, CPE colonization was established as the reference point for univariate and multivariate analysis of risk factors of CPE acquisition. The reference for each characteristic was based on age < 70 years, female, and other variables being “no” or “none.” The confidence interval (CI) was set at 95%, and *p*-value < 0.05 was considered to be statistically significant. All data analyses were performed using the Jamovi program (version 1.6.23; https://www.jamovi.org) and R Core Team (2020). *R: A Language and environment for statistical computing.* (Version 4.0). (https://cran.r-project.org; R packages retrieved MRAN snapshot 2020-08-24) (accessed on 2 September 2022).

## 5. Conclusions

This study provides insights into the epidemiological and clinical characteristics of CPE infection and the risk factors associated with CPE acquisition. Our data reveal a predominant gene of KPC-producing *Klebsiella pneumoniae* strains and a high proportion of patients with hospital history at other hospitals within 3 months in the CPE colonization group. CPE acquisition was significantly associated with ICU stay, open wounds, holding urinary or central catheters, gastrointestinal tubes, and antibiotic treatment. Therefore, active screening should be implemented to detect CPE on admission and periodically for high-risk patients, including inpatients with a hospitalization history within 3 months.

## Figures and Tables

**Table 1 antibiotics-12-00759-t001:** Characteristics of patients with CPE infection according to colonization and acquisition (*n* = 77).

Characteristic		Total (*n* = 77)		Colonization (*n* = 51)		Acquisition (*n* = 26)		*p*-Value
Age (median, interquartile range)		77 (67.0–84.0)		80.0 (71.5–85.0)		71.5 (65.3–79.8)		
	<70 years	23	(29.9)	11	(21.6)	12	(46.2)	0.036
	≥70 years	54	(70.1)	40	(78.4)	14	(53.8)	
Sex								
	Male	41	(53.2)	24	(47.1)	17	(65.4)	0.152
	Female	36	(46.8)	27	(52.9)	9	(34.6)	
Hospitalization history at other hospitals within 3 months								
	Yes	54	(70.1)	41	(80.4)	13	(50.0)	0.009
	No	23	(29.9)	10	(19.6)	13	(50.0)	
Admission route								
	ER	33	(42.9)	25	(49.0)	8	(30.8)	0.026
	Ward	35	(45.5)	24	(47.1)	11	(42.3)	0.692
	ICU	9	(11.7)	2	(3.9)	7	(26.9)	0.003
Comorbidity								
	Yes	70	(90.9)	47	(92.2)	23	(88.5)	0.68
	No	7	(9.1)	4	(7.8)	3	(11.5)	
Type of specimen								
	Rectal swab (stool)	52	(67.5)	37	(72.5)	15	(57.7)	0.21
	Ect ^a^	25	(32.5)	14	(27.5)	11	(42.3)	
Bacterial species								
	*Klebsiella pneumoniae*	60	(77.9)	42	(82.4)	18	(69.2)	0.07
	*Escherichia coli*	14	(18.2)	9	(17.6)	5	(19.2)	
	*Citrobacter freundii*	2	(2.6)	0	(0.0)	2	(7.7)	
	*Citrobacter amalonaticus*	1	(1.3)	0	(0.0)	1	(3.8)	
Carbapenemase-gene								
	KPC	69	(89.6)	48	(94.1)	21	(80.8)	0.07
	NDM	5	(6.5)	2	(3.9)	3	(11.5)	
	OXA-48	1	(1.3)	1	(2.0)	0	(0.0)	
	NDM and OXA-48	2	(2.6)	0	(0.0)	2	(7.7)	

Data are presented as *n* (%). CPE, carbapenemase-producing *Enterobacteriaceae*; ER, emergency room; ICU, intensive care unit; KPC, *Klebsiella pneumoniae* carbapenemase; NDM, New Delhi metallo beta-lactamase; OXA-48, oxacillinase-48. ^a^ Ect (sputum, urine, wound (pus), body fluid, and blood).

**Table 2 antibiotics-12-00759-t002:** Differences in clinical risk factors associated with CPE colonization and acquisition.

Characteristic		Total (*n* = 77)		Colonization (*n* = 51)		Acquisition (*n* = 26)		*p*-Value
Days from admission to CPE-positive sample collected (mean ± SD)		6.0 ± 13.0		0.5 ± 0.7		16.8 ± 18.1		<0.001 ^a^
Treatment in ICU								
	Yes	31	(40.3)	10	(19.6)	21	(80.8)	<0.001
	No	46	(59.7)	41	(80.4)	5	(19.2)	
Open wound								
	Yes	39	(50.6)	22	(43.1)	17	(65.4)	0.092
	No	38	(49.4)	29	(56.9)	9	(34.6)	
Surgery ^b^								
	Yes	15	(19.5)	8	(15.7)	7	(26.9)	0.361
	No	62	(80.5)	43	(84.3)	19	(73.1)	
Invasive device ^c^								
	None	16	(20.8)	14	(27.5)	2	(7.7)	0.070
	Urinary catheter	52	(67.5)	30	(58.8)	22	(84.6)	0.038
	Central catheter	33	(42.9)	13	(25.5)	20	(76.9)	<0.001
	Gastrointestinal tube	22	(28.6)	9	(17.6)	13	(50.0)	0.007
	Drainage tube	9	(11.7)	6	(11.8)	3	(11.5)	1
	Mechanical ventilation	10	(13.0)	3	(5.9)	7	(26.9)	0.026
Antibiotic exposure ^c^								
	None	12	(15.6)	11	(21.6)	1	(3.8)	0.051
	Carbapenems	28	(36.4)	14	(27.5)	14	(53.8)	0.027
	Penicillins	28	(36.4)	17	(33.3)	11	(42.3)	0.463
	Cephalosporins	42	(54.5)	26	(51.0)	16	(61.5)	0.470
	Glycopeptides	20	(26.0)	8	(15.7)	12	(46.2)	0.006
	Fluoroquinolones	42	(54.5)	27	(52.9)	15	(57.7)	0.810
	Vancomycin	10	(13.0)	5	(9.8)	5	(19.2)	0.292

Data are presented as mean ± SD or *n* (%). CPE, carbapenemase-producing *Enterobacteriaceae*; ICU, intensive care unit. ^a^ *t*-test, ^b^ Excluding simple surgery, ^c^ Multiple responses.

**Table 3 antibiotics-12-00759-t003:** Univariate and multivariate analyses of risk factors for CPE acquisition after hospitalization ^a^ (*n* = 77).

Characteristic		Univariate Analysis			Multivariate Analysis		
		OR (CI)		*p*-Value	aOR (CI)		*p*-Value
Age							
	≥70 years (ref. <70 years)	0.32	(0.012–0.89)	0.029	0.51	(0.16–1.63)	0.257
Sex							
	Male (ref. Female)	2.13	(0.80–5.65)	0.131	1.63	(0.53–5.02)	0.396
Hospitalization history of other hospitals within 3 months							
	Yes (ref. No)	0.24	(0.09–0.69)	0.007	0.32	(0.10–0.98)	0.047
Admission route (ref. ER)							
	Ward	1.43	(0.49–4.17)	0.510	1.54	(0.49–4.87)	0.462
	ICU	10.94	(1.88–63.68)	0.008	7.14	(1.09–46.80)	0.040
Comorbidity							
	Yes (ref. None)	0.65	(0.13–3.16)	0.596	0.77	(0.10–6.08)	0.800
Treatment in ICU							
	Yes (ref. None)	17.22	(5.21–56.91)	<0.001	46.72	(5.08–430.09)	<0.001
Open wound							
	Yes (ref. No)	2.49	(0.94–6.63)	0.068	5.99	(1.52–23.60)	0.011
Surgery ^b^							
	Yes (ref. No)	1.98	(0.63–6.25)	0.244	2.84	(0.76–10.66)	0.121
Invasive device (ref. None)							
	Urinary catheter	3.85	(1.16–12.81)	0.028	4.56	(1.13–18.39)	0.033
	Central catheter	9.74	(3.22–29.52)	<0.001	11.94	(2.98–47.93)	<0.001
	Gastrointestinal tube	4.67	(1.63–13.38)	0.004	12.70	(2.61–61.84)	0.002
	Drainage tube	0.98	(0.22–4.27)	0.977	1.03	(0.21–5.18)	0.968
	Mechanical ventilation	5.90	(1.38–25.21)	0.017	4.86	(0.86–27.49)	0.074
Antibiotic exposure (ref. None)							
	Carbapenems	3.08	(1.15–8.27)	0.025	4.88	(1.40–16.95)	0.013
	Glycopeptides	4.61	(1.57–13.55)	0.006	5.13	(1.46–18.00)	0.011
	Penicillins	1.47	(0.56–3.88)	0.772	1.00	(0.32–3.14)	0.998
	Cephalosporins	1.54	(0.59–4.03)	0.380	1.42	(0.47–4.26)	0.536
	Fluoroquinolones	1.21	(0.47–3.14)	0.692	1.51	(0.49–4.71)	0.474
	Vancomycin	2.19	(0.57–8.39)	0.252	3.89	(0.80–18.89)	0.092

CPE, carbapenemase-producing *Enterobacteriaceae*; ER, emergency room; ICU, intensive care unit; OR, odds ratio; aOR, adjusted odds ratio; CI, confidence interval. ^a^ Reference group = CPE colonization on admission, ^b^ Excluding simple surgery.

## Data Availability

Not applicable.

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
