# Peer review of "Epidemiology and Risk Factors of Carbapenemase-Producing Enterobacteriaceae Acquisition and Colonization at a Korean Hospital over 1 Year"

_antibiotics, 2023, doi:10.3390/antibiotics12040759_

Round 1
Reviewer 1 Report
The study is important in current scenario of AMR.
and such studies are highly required but following points for consideration by authors
1. Language check highly required.
2. Italicize the names of microbes speciall K.pneumoniae
3. Introduction to be improved
4. Sample size is too low
5
Reviewer 2 Report
Bacteria names should be checked according to the microorganism name spelling rules (in the abstract section).
Table 1 fonts should be checked.
In general, no information is given about microbiological methods. For example;
Identification of Enterobacteriaceae, antimicrobial susceptibility test, which method and/or which system was used? Even if the article is about the analysis of the results, it would be good to mention which methods are used in the method section and the main materials in the material section.
It would be good to provide Carbapenemase Class information (A,B,D)
Carbapenem MIC50/90 value and MIC range of resistant K. pneumoniae strains should be given.
In the discussion section, the distribution of resistant strains by years can be compared with other studies.
Reviewer 3 Report
The authors have elucidated the risk factors and characteristics of patients with CRE in a hospital setting. Although the study is interesting, the presentation and analysis can be significantly improved.
The most obvious issue is the poor quality of the text in terms of both grammatical and spelling errors and poor structuring of sentences. This issue makes it a challenging manuscript to read and can lead to grave misinterpretations.
The second issue is lack of a non-CRE group. I suggest the authors add a this matched group from the same hospital that was not colonized and did not acquire CRE during their stay. Then some of the claims (e.g. glycopeptide use being a risk factor for CPE) can be solidified.
Please take these suggestions as constructive feedback.
Author Response
Please refer to the attached file.
Thank you。

Reviewer 4 Report
Dear authors,
I have a few remarks and suggestions as follows:
1. The title is too long. Please edit it.
2. Section "Materials and Methods": Please add a brief description regarding the species identification of CRE, as well as the method for carbapenemase genes detection.
3. Please italicize all bacterial species names (e.g. line 248).
4. The syntax needs improvement in places.
Author Response

(The authors gave the same response as above.)

Round 2
Reviewer 3 Report
Thank you for addressing my concerns.